# Relationship between the strength of craving as assessed by the Tobacco Craving Index and success of quitting smoking in Japanese smoking cessation therapy

Chie Taniguchi[1,2]*, Hideo Saka[3,4], Isao Oze[5], Sumie Nakamura[6], Yasuhiro Nozaki[7,8], Hideo Tanaka[9]

1 College of Nursing, Aichi Medical University, Nagakute-city, Aichi, Japan, 2 Department of Clinical Research Center, National Hospital Organization Nagoya Medical Center, Nagoya-city, Aichi, Japan, 3 Department of Respiratory Medicine, Matsunami General Hospital, Hashima-gun, Gifu, Japan, 4 Department of Respiratory Medicine, National Hospital Organization Nagoya Medical Center, Nagoya-city, Aichi, Japan, 5 Division of Cancer Epidemiology and Prevention, Department of Preventive Medicine, Aichi Cancer Center Research Institute, Nagoya-city, Aichi, Japan, 6 Department of Nursing, National Hospital Organization Nagoya Medical Center, Nagoya-city, Aichi, Japan, 7 Department of Respiratory Medicine, Japan Community Healthcare Organization Chukyo Hospital, Nagoya-city, Aichi, Japan, 8 Department of Respiratory Medicine, Tokoname Municipal Hospital, Tokoname-city, Aichi, Japan, 9 Fujiidera Public Health Center of Osaka Prefecture, Fujidera-city, Osaka, Japan

* c-taniguchi@aichi-med-u.ac.jp

**Data Availability Statement:** All relevant data are within the manuscript and its Supporting information files.

## Abstract

### Background

We previously developed the Tobacco Craving Index (TCI) to assess craving of smokers. In the present study, we validated the relationship between the TCI grade over the 5 sessions of Japanese smoking cessation therapy (SCT) and success of quitting smoking among 889 Japanese patients.

### Methods

The Japanese SCT consists of 5 sessions of SCT (first session and sessions 2, 4, 8 and 12 weeks later). In the TCI questionnaire, patients are asked to rate their strength of craving and frequency of craving, each on a four-point Likert scale. Patients are classified into one of four grades based on their responses (0, I, II, III, with III indicating severe craving). The TCI questionnaire was administered to each participant at each session of the SCT. This study included participants of Japanese SCT who answered the TCI at the first session of the SCT at five Japanese smoking cessation clinics. Patients who dropped out of the SCT from the second to the fifth sessions were considered to have failed smoking cessation. To elucidate how much the TCI grade predicts smoking status at the last session, we performed multivariate logistic regression analysis with adjustment for confounding factors.

### Results

Participants who had higher TCI grade(III) in the 2nd through 5th sessions showed significantly lower probability for success of quitting smoking than those who had lower TCI

**Funding:** This work was supported by a MEXT KAKENHI Grant-in-Aid for Young Scientist, Grant Number 19K19582 who received C.T. (https://www.jsps.go.jp/j-grantsinaid/) The funders had no role in study design, data collection and analysis, decision to publish, or preparation of the manuscript.

**Competing interests:** The authors have declared that no competing interests exist.

grades(0 or I) (adjusted odds ratio: 2nd session: 0.30, 3rd session: 0.15, 4th session: 0.06, 5th session: 0.02).

## Conclusions

We validated the usefulness of the TCI grade for assessing probability of quitting smoking by using a large number of smoking cessation settings.

## Introduction

Craving is an intense and prolonged desire and is considered as a central feature of addiction [1, 2]. Smokers feel nicotine craving in various situations during their daily life, and they often have difficulty in quitting smoking because of their craving [3]. Previous studies suggested that a higher craving level among smokers is associated with failure of attempt to quitting smoking [4, 5] and increases the risk of relapse [6, 7]. However, many studies that examined craving and smoking cessation had a small sample size [8–10].

We previously developed a new craving index named the Tobacco Craving Index (TCI) which consists of two axes: the strength and frequency of craving [11]. Although the TCI consists of only two axes, it showed almost the same sensitivity and specificity in predicting the success of quitting smoking as the brief form of the Questionnaire of Smoking Urges (QSU-brief) in a Japanese smoking cessation therapy (SCT) setting in a limited sample size of 85 smokers [11]. Development of an easy-to-use and reliable indicator of craving is important in clinical settings. Therefore, validation of the usefulness of the TCI is needed to assess the relationship between the TCI score and success of smoking cessation in a larger number of participants in the SCT.

The aim of this study was to validate the association between the TCI grade over the five sessions of SCT and success of quitting smoking in a large number of Japanese SCT participants.

## Methods

### Settings and participants

Japanese SCT has been covered by health insurance starting from 2006. The SCT consists of a total of 5 sessions over 12 weeks. Patients receive smoking cessation treatment 5 times over 12 weeks: the first session and 2, 4, 8, and 12 weeks after the first session. Patients set a quit date one week after the first visit. At each session, patients receive medical treatment for smoking cessation by a physician for approximately 15 minutes, and behavioral counseling by nurses (15–30 minutes). Details of the Japanese SCT are available elsewhere [12, 13].

We performed a multi-institutional study to monitor the effect of SCT administered at five Japanese hospitals (Nagoya Medical Center, Aichi Cancer Center, Chukyo Hospital, Kinki—Chuo Chest Medical Center, and Kitazato Research Hospital) between October 2008 and June 2014. During this period, 1,381 patients received SCT at the above-mentioned five hospitals. Of these, 1,324 patients provided written informed consent to receive the Japanese SCT. The TCI questionnaire was administered to each patient at every session of the SCT starting in October 2009. The subjects of the present study were 889 patients who gave written informed consent to receive Japanese SCT and who answered the TCI questionnaire in the first session of the SCT between October 2009 and June 2014. In the SCT, patients were given a prescription for either varenicline or nicotine patches.

This study was approved by the Institutional Review Board of Nagoya Medical Center.

## Data collection

Demographic data including age, gender, presence of a cohabitant(s), having experience of quitting smoking and Brinkman Index were obtained from self-report questionnaires at the first session of the SCT. A cohabitant was defined as a person living together with the SCT participant. We assessed the strength of nicotine dependence with Fagerström Test for Nicotine Dependence (FTND). In this test, scores range from 0 to 10, and we defined seven and over as indicating high nicotine dependence [14]. We screened for depression at the beginning of each session by administering the Center for Epidemiologic Studies Depression Scale (CES-D). The CES-D consists of 20 items, and total scores range from 0 to 60, with scores $\geq 16$ indicating possible clinical depression [15, 16]. Participants' present motivation and self-efficacy for quitting smoking were asked during nurses' counseling at each session on a scale from 0 to 100%.

## Definition of success of quitting smoking

We defined success of quitting smoking as the condition that participants had quit smoking for at least 2 weeks before the last (5th) session, which was verified by the carbon monoxide concentration in expired air ($\leq 7$ ppm). Those who dropped out of the SCT from the second to the fifth session were defined as having failed quitting smoking.

## Craving scale

The TCI is a useful indicator to assess tobacco craving in patients undergoing SCT that we created [11]. The TCI questionnaire consists of two items, and patients were asked to rate each item on a four-point Likert scale. The first item asks about the strength of craving (0: I feel no craving for smoking anymore, 1: I feel a need to put something in my mouth to cope with the craving, 2: I need endurance to cope with the craving. 3: I can hardly continue to stop smoking because of a strong craving), and the second item asks about the frequency of craving (0: 0 times per day, 1: < 1 time per day, 2: 1~3 times per day, 3: $\geq 4$ times per day). Patients were classified into one of four TCI grades (0, I, II or III, with III indicating severe craving) depending on their answers to the two questions. Details are shown in S1 Fig.

## Statistical methods

We analyzed the relationships between the participants' characteristics and TCI grade at the first session. Mann-Whitney U test and Kruskal-Wallis test were used in this analysis. Comparison of the success of quit rate at the end of SCT according to TCI grade at each session was performed. We then described the distribution of TCI grade through the 5 sessions of SCT in patients who succeeded in quitting smoking by the last session of the SCT (Success group) and in patients who did not succeed in quitting smoking by the last session (Failure group). At each session, to elucidate how much the TCI grade predicts smoking status at the last session, we performed multivariate logistic regression analysis with adjustment for variables that showed statistical significance in Table 1: gender (female/ male), age (continuous variable), having a present illness (absence/ presence), FTND (< 7/ $\geq$7), CES-D at the first session (< 16/ $\geq$16), prescription (nicotine patches / varenicline), motivation for quitting smoking at the 1st session (continuous variable), and self-efficacy for quitting smoking at the 1st session (continuous variable).

Table 1. Characteristics of the study subjects according to the TCI grade at the 1st session of SCT.

| | | TCI grade | | | | | | | | | | | p-value | Total | |
| | | Grade 0 | | | Grade I | | | Grade II | | | Grade III | | | | | |
| | | n | % | Mean (SD) | n | % | Mean (SD) | n | % | Mean (SD) | n | % | Mean (SD) | | | |
| Gender | Male | 18 | 90.0 | | 114 | 67.1 | | 190 | 70.9 | | 284 | 65.9 | | 0.183 | 606 | 68.2 |
| | Female | 2 | 10.0 | | 56 | 32.9 | | 78 | 29.1 | | 147 | 34.1 | | | 283 | 31.8 |
| Age (y)[a] | | | | 62.9 (14.8) | | | 55.3 (14.3) | | | 53.4 (14.2) | | | 51 (13.9) | <0.001 | 52.8 | 14.3 |
| Having experience of quitting smoking | Absence | 7 | 35.0 | | 36 | 21.3 | | 54 | 20.6 | | 111 | 25.9 | | 0.241 | 208 | 23.6 |
| | Presence | 13 | 65.0 | | 133 | 78.7 | | 208 | 79.4 | | 318 | 74.1 | | | 672 | 76.4 |
| Cohabitant* | Absence | 9 | 45.0 | | 66 | 60.7 | | 85 | 32.2 | | 153 | 36.2 | | 0.702 | 313 | 35.8 |
| | Presence | 11 | 55.0 | | 102 | 39.3 | | 179 | 67.8 | | 270 | 63.8 | | | 562 | 64.2 |
| Having a present illness | Absence | 2 | 10.0 | | 17 | 10.0 | | 44 | 16.4 | | 75 | 17.4 | | 0.041 | 138 | 15.5 |
| | Presence | 18 | 90.0 | | 153 | 90.0 | | 224 | 83.6 | | 356 | 82.6 | | | 751 | 84.5 |
| Fagerström test for nicotine dependence | <7 | 15 | 83.3 | | 133 | 78.7 | | 190 | 73.1 | | 205 | 50.1 | | <0.001 | 543 | 63.4 |
| | ≥7 | 3 | 16.7 | | 36 | 21.3 | | 70 | 26.9 | | 204 | 49.9 | | | 313 | 36.6 |
| CES-D at the 1st session | <16 | 17 | 72.6 | | 122 | 72.6 | | 188 | 72.6 | | 240 | 65.0 | | 0.015 | 567 | 69.5 |
| | ≥16 | 3 | 27.4 | | 46 | 27.4 | | 71 | 27.4 | | 129 | 35.0 | | | 249 | 30.5 |
| Prescription | Nicotine Patch | 4 | 20.0 | | 28 | 16.5 | | 36 | 13.4 | | 92 | 21.4 | | 0.043 | 160 | 18.0 |
| | Varenicline | 16 | 80.0 | | 142 | 83.5 | | 232 | 86.6 | | 339 | 78.7 | | | 729 | 82.0 |
| Motivation for quitting smoking at the 1st session [a] | | | | 85.5 (21.0) | | | 82.2 (23.0) | | | 83.3 (20.5) | | | 77.3 (25.7) | 0.035 | 80.3 | 23.8 |
| Self-efficacy for quitting smoking at the 1st session [a] | | | | 71.6 (31.5) | | | 59.1 (28.8) | | | 60.9 (28.8) | | | 50.7 (28.7) | <0.001 | 55.9 | 29.3 |

Mann-Whitney U test

[a]: Kruskal-Wallis test

*A cohabitant was defined as a person living together with the SCT participant.

## Results

Table 1 summarizes the characteristics of the 889 study subjects according to TCI grade at the first session of the SCT. The study participants ranged in age from 17 to 88 years and their mean age was 52.8±14.3 (standard deviation) years. The mean age gradually decreased as the baseline TCI grade increased (p<0.001). Thirty-six percent (313/889) of the participants had high nicotine dependence. As the TCI grade increased, the proportion of participants with high nicotine dependence increased (p<0.001). About thirty percent (249//889) of the participants had clinical depression according to the CES-D. Patients who had a TCI grade of III at the first session were significantly more depressed, less motivated, and had less self-efficacy than those who had a TCI grade of 0, I, or II.

More than 80% of participants who had TCI grade of 0 or I at each session were successful in quitting smoking at the last session (Fig 1). As the TCI grade increased, the proportion of participants who failed in quitting smoking increased, and this trend became more significant through the 5 sessions (1st session: p = 0.012 and 2nd to 5th session: p<0.001) (Fig 1).

Among the 889 participants, 359 participants succeeded in quitting smoking by the last session of the SCT and were placed in the Success group. In the Success group (n = 359), the distribution of TCI grades shifted towards a lower grade through the 5 sessions of the SCT (Fig 2a). In contrast, the proportion of participants with high TCI grade (II or III) in the Failure

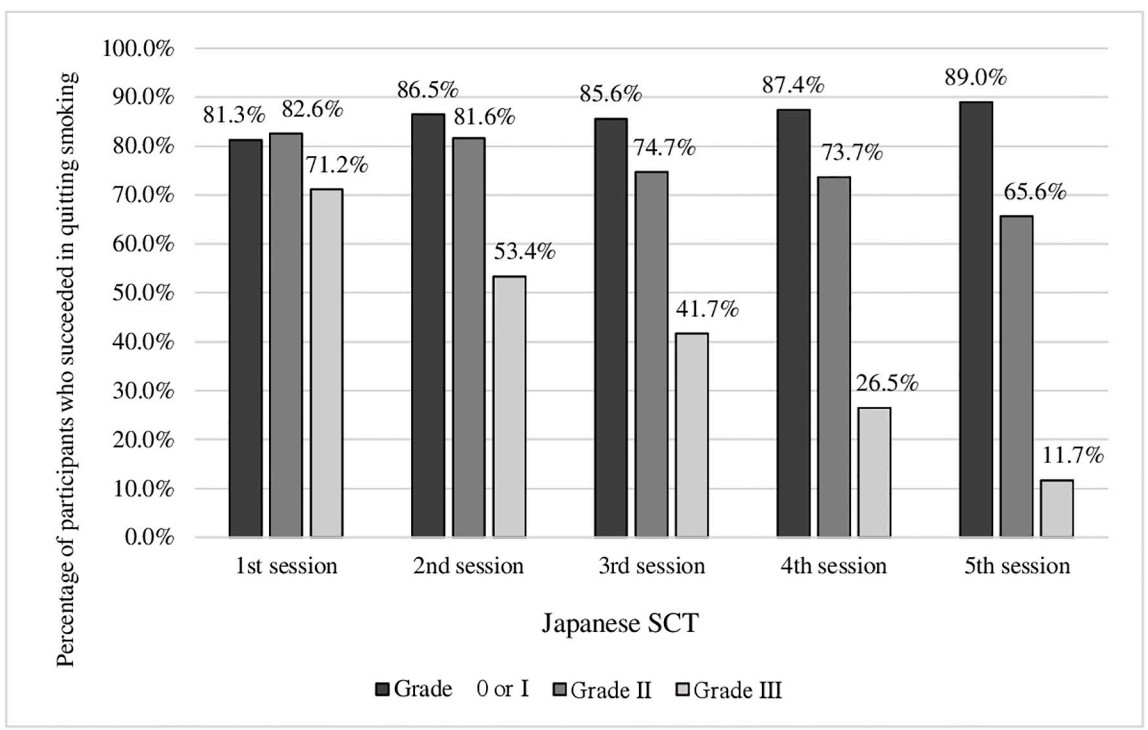

**Fig 1. Success rate of quitting smoking at the end of the SCT according to the TCI grade at each session of Japanese SCT.**

group (n = 530) at the first, third and fifth sessions were 80.6%, 50.8% and 63.8%, respectively, which showed a u-shaped proportion curve with a nadir at the third session (Fig 2b).

We performed logistic regression analyses to identify the associations between TCI grade at the first to fifth sessions and the success of quitting smoking at the fifth session, while not adjusting and adjusting for confounding factors (Table 2). Compared with participants whose TCI grade was 0 or I, those who had TCI grade III had significantly lower probability of success of quitting smoking at each session (adjusted odds ratio: 2nd session: 0.30, 3rd session: 0.15, 4th session: 0.06, 5th session: 0.02). Compared with participants whose TCI grade was 0 or I, those who had TCI grade II had significantly lower probability for success of quitting smoking from the 3rd to 5th session of the SCT (adjusted odds ratio: 3rd through 5th session: 0.49, 0.38 and 0.23). There were no significant associations between the success of quitting smoking and TCI grade II or III at the 1st session (Table 2).

## Discussion

We validated that the Japanese SCT participants who had higher TCI grade in the 2nd through 5th sessions showed significantly lower probability for success of quitting smoking than those who had lower TCI grades.

We previously examined how closely TCI grade is associated with success of quitting smoking among 85 individuals who received Japanese SCT compared with the QSU-brief (there was no overlap of participants in the previous study and the present study) [11]. The results suggested that the TCI, as well as QSU-brief, can be used as a predictive tool for success of quitting smoking. Previous studies indicated that high nicotine dependence and depression are associated with strong tobacco craving [17, 18]. Also, Reese et al. reported that decreased self-efficacy for abstinence is linked to increased craving [19]. The present study showed that

a) Success group (n=359)

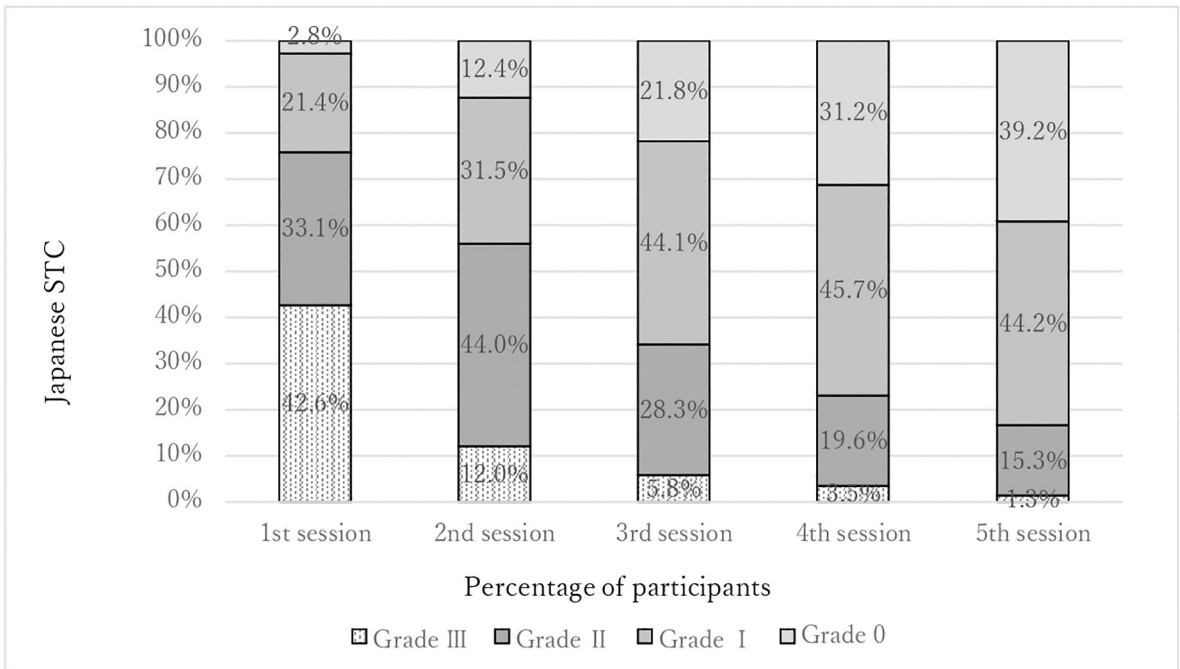

b) Failure group (n=530)

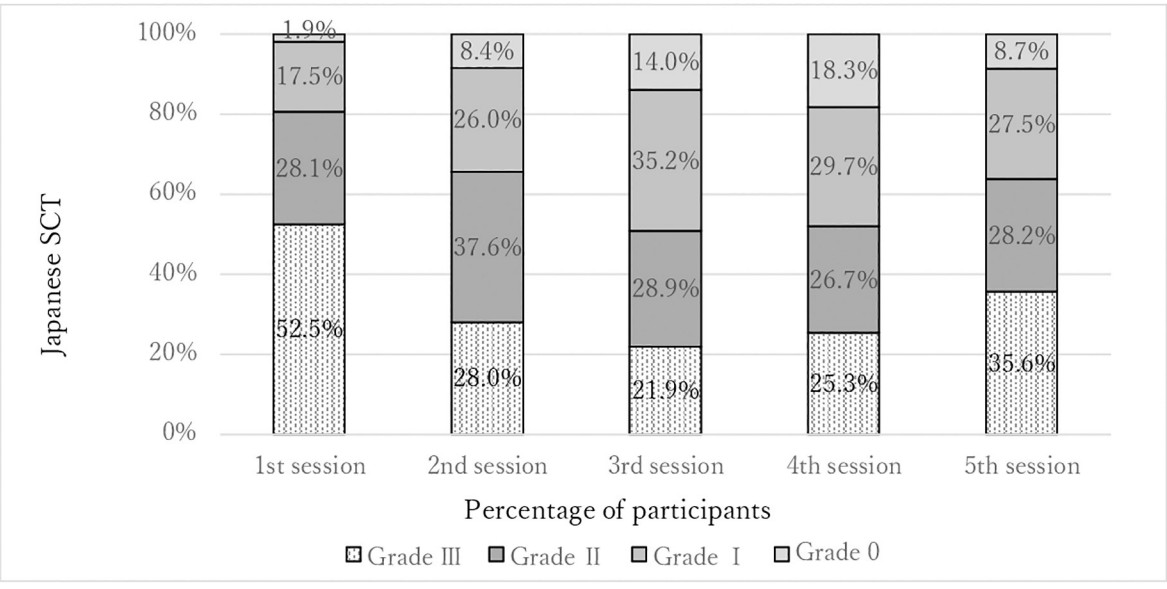

**Fig 2.** (a) Changes in the TCI grade through the 5 sessions of Japanese SCT in the Success group. (b) Changes in the TCI grade through the 5 sessions of Japanese SCT in the Failure group.

**Table 2. Odds ratio of TCI grade for success of quitting smoking at the last session of Japanese SCT.**

|  | TCI grade | Unadjusted odds ratio | [95% CI] | p for trend | Adjusted odds ratio* | [95% CI] | p for trend |
|---|---|---|---|---|---|---|---|
| 1st session | Grade 0, I | 1.00 |  | 0.021 | 1.00 |  | 0.307 |
|  | Grade II | 1.09 | [0.57–2.10] |  | 1.13 | [0.56–2.28] |  |
|  | Grade III | 0.57 | [0.32–1.00] |  | 0.74 | [0.38–1.44] |  |
| 2nd session | Grade 0, I | 1.00 |  | <0.001 | 1.00 |  | <0.001 |
|  | Grade II | 0.91 | [0.70–1.19] |  | 0.92 | [0.55–1.55] |  |
|  | Grade III | 0.34 | [0.24–0.48] |  | 0.30 | [0.17–0.56] |  |
| 3rd session | Grade 0, I | 1.00 |  | <0.001 | 1.00 |  | <0.001 |
|  | Grade II | 0.50 | [0.33–0.77] |  | 0.49 | [0.30–0.80] |  |
|  | Grade III | 0.12 | [0.07–0.21] |  | 0.15 | [0.08–0.29] |  |
| 4th session | Grade 0, I | 1.00 |  | <0.001 | 1.00 |  | <0.001 |
|  | Grade II | 0.40 | [0.25–0.66] |  | 0.38 | [0.22–0.64] |  |
|  | Grade III | 0.05 | [0.03–0.09] |  | 0.06 | [0.03–0.13] |  |
| 5th session | Grade 0, I | 1.00 |  | <0.001 | 1.00 |  | <0.001 |
|  | Grade II | 0.24 | [0.15–0.38] |  | 0.23 | [0.14–0.40] |  |
|  | Grade III | 0.02 | [0.01–0.04] |  | 0.02 | [0.01–0.05] |  |

*Adjusted for gender (female/ male), age (continuous variable), having a present illness (absence/ presence), FTND ($< 7/ \geq 7$), CES-D at the first session ($< 16/ \geq 16$), prescription (nicotine patches/ varenicline), motivation for quitting smoking at the 1st session (continuous variable), and self-efficacy for quitting smoking at the 1st session (continuous variable).

higher TCI grade was significantly associated with failure of quitting smoking after adjusting for gender, age, FTND, CES-D, type of prescription, motivation for quitting smoking, self-efficacy for quitting smoking and having a present illness. The present results together with our previous findings [11] demonstrated that TCI grading is an independent predictor for success of quitting smoking in smoking cessation settings. Therefore, we think that the TCI is a useful indicator for monitoring the efficacy of success of quitting smoking during smoking cessation treatment.

Our study also found that there was no significant association between the TCI grade at the 1st session and success of quitting smoking. Wray et al. performed a systematic review of studies on craving and smoking cessation, and found that studies showed significant associations when using only post-cessation but not pre-cessation assessments [20]. Measurements of pre-cessation craving may be subject to floor effects which would make it difficult to detect significant relationships between craving and outcome.

Among the patients who succeeded in quitting smoking in the present study, the distribution of TCI grade shifted towards a lower grade through the 5 sessions of the SCT. Shiffman and Jarvik (1976) examined the relationship between two weeks' abstinence and craving score and found that the linear and quadratic trends of the curve were significantly different between the success and failure groups [21]. The craving score of participants who succeeded in quitting smoking dropped sharply compared with the craving score of those who failed to quit smoking [21] as in our study. Shiffman and Jarvik (1976) examined only 2 weeks in smoking cessation treatment. Few studies have investigated the change in craving over several months according to the smoking status. The duration of observation in previous studies that examined the relationship between change in craving level and success of quitting smoking was less than the first seven days in most studies [22–24]. Zuo et al. (2017) examined the associations of change in negative affect and craving during a 44-day period of smoking abstinence with cessation outcome at 3 months and 1 year [25]. They demonstrated that greater declines in anxiety, depression and anger symptoms over the first 44 days of smoking cessation were

predictive of higher odds of abstinence at 3 months and at 1 year [25]. They discussed negative affect and craving to be two major components of tobacco withdrawal that each uniquely predict failure of quitting smoking. Our findings together with these results suggest that controlling tobacco craving by appropriate medication and counselling that contains strategies for coping with urges can reduce the risk of relapse of smoking.

A strength of the present study was that we examined the relationship between the TCI and success of quitting smoking in a large number of participants. To assess the relationship, we adjusted several important confounding factors on smoking behaviors in multivariate analysis. On the other hand, a potential limitation was that our study had a short observation period of 3 months during SCT. Therefore, the association between the TCI and long-term smoking cessation needs to be investigated in the future.

In conclusion, our study demonstrated that higher TCI grade is associated with failure of quitting smoking in a large number of Japanese SCT participants. As the TCI questionnaire consists of only two questions, patients can easily fill it out. Therefore, we think that it is a useful questionnaire that can be administered in various clinical settings for the assessment of tobacco craving. These findings indicate that monitoring the TCI of participants during smoking cessation intervention might be useful when considering appropriate medication and counseling methods. Further studies are needed to validate the TCI in countries other than Japan.

## Supporting information

**S1 Fig. Tobacco Craving Index (TCI).**
(DOCX)

## Acknowledgments

The authors wish to acknowledge Dr. Kohta Suzuki, Professor of Department of Health and Psychosocial Medicine Aichi Medical University School of Medicine for his help in interpreting the significance of the results of this study. We also thank all medical staffs that supported this smoking cessation therapy at Nagoya Medical Center, Aichi Cancer Center, Chukyo Hospital, Kinki—Chuo Chest Medical Center, and Kitazato Research Hospital.

## Author Contributions

**Conceptualization:** Chie Taniguchi, Hideo Tanaka.

**Data curation:** Chie Taniguchi, Isao Oze, Sumie Nakamura, Yasuhiro Nozaki, Hideo Tanaka.

**Formal analysis:** Chie Taniguchi, Isao Oze, Hideo Tanaka.

**Funding acquisition:** Chie Taniguchi.

**Investigation:** Chie Taniguchi, Isao Oze, Sumie Nakamura, Yasuhiro Nozaki, Hideo Tanaka.

**Methodology:** Chie Taniguchi, Hideo Tanaka.

**Project administration:** Hideo Saka.

**Resources:** Chie Taniguchi, Isao Oze, Sumie Nakamura, Yasuhiro Nozaki.

**Software:** Chie Taniguchi.

**Validation:** Chie Taniguchi, Isao Oze, Sumie Nakamura, Yasuhiro Nozaki.

**Visualization:** Chie Taniguchi.

Writing – **original draft:** Chie Taniguchi.

Writing – **review & editing:** Chie Taniguchi, Hideo Saka, Isao Oze, Sumie Nakamura, Yasu-hiro Nozaki, Hideo Tanaka.

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
