## [Decision Letter · Decision Letter 0]

21 Oct 2020

PONE-D-20-25851

Relationship between the strength of craving as assessed by the Tobacco Craving Index (TCI) and success of quitting smoking in Japanese smoking cessation therapy

PLOS ONE

Dear Dr. Taniguchi,

Thank you for submitting your manuscript to PLOS ONE. After careful consideration, we feel that it has merit but does not fully meet PLOS ONE’s publication criteria as it currently stands. Therefore, we invite you to submit a revised version of the manuscript that addresses the points raised during the review process.

We look forward to receiving your revised manuscript.

Kind regards,

Neal Doran

Academic Editor

PLOS ONE

Journal Requirements:

2. Please include additional information regarding the survey or questionnaire used in the study and ensure that you have provided sufficient details that others could replicate the analyses.

For instance, if you developed a questionnaire as part of this study and it is not under a copyright more restrictive than CC-BY, please include a copy, in both the original language and English, as Supporting Information.

Reviewers' comments:

Reviewer's Responses to Questions

**Comments to the Author**

1. Is the manuscript technically sound, and do the data support the conclusions?

Reviewer #1: Yes

Reviewer #2: Yes

2. Has the statistical analysis been performed appropriately and rigorously? 

Reviewer #1: Yes

Reviewer #2: Yes

3. Have the authors made all data underlying the findings in their manuscript fully available?

Reviewer #1: Yes

Reviewer #2: Yes

4. Is the manuscript presented in an intelligible fashion and written in standard English?

Reviewer #1: Yes

Reviewer #2: Yes

5. Review Comments to the Author

Reviewer #1: In this manuscript entitled “Relationship between the strength of craving as assessed by the Tobacco Craving Index (TCI) and success of quitting smoking in Japanese smoking cessation therapy”, the authors evaluated correlations of the unique craving score of smoking cessation, TCI and success of smoking cessation. They reported that higher TCI grade in 2nd to 5th sessions were associated with failure of quitting smoking at the 5th session (however 1st was not). I have some comments to be addressed.

Major comments:

1. Please clarify the benefit of using the TCI instead of the FTND-based general craving score in this study (e.g. easy to use, less time to fill them out…?).

2. How many patients received smoking cessation medications as nicotine patch or varenicline? Since these medications can ameliorate the cravings, low prescription rate may affect the results.

3. Table 1. What is “cohabiter”? Please clarify and explain it.

4. Table 1, last 2 rows. It seems that 2 tables were mixed, “N (%)” and “mean, SD”, very confusing. Please separate them or change the style.

5. How do you think we can apply these results into the smoking cessation program? Should we aggressively intervene cravings at 2nd to 4th sessions for smoking cessation success?

Minor comments:

1. Please remove abbreviation “(TCI)” from the title.

2. In conclusion, you cannot say definitive things from the study. Change the last sentence as “These findings indicate that … smoking cessation interventions might be useful …”.

Reviewer #2: Methods Settings and participants

・The authors mention ‘We selected 889 participants who answered the TCI questionnaire in first session of the SCT.’ I wonder how to select this study population. Did the authors offer Tobacco Craving Index questionnaire to all participants in the smoking cessation therapy? Did these 889 participants who answered the TCI questionnaire in first session of the SCT complete all 5 sessions of Japanese smoking cessation therapy?

・Please describe medical treatment in the smoking cessation therapy, that is, transdermal nicotine patches or varenicline. Prescribed pharmacotherapy might be associated with success of quitting smoking as a confounding factor.

Discussion

・The authors should mention that advantage of their Tobacco Craving Index questionnaire compared with other tools to evaluate nicotine craving. In addition, I think their Tobacco Craving Index questionnaire should be validated in other countries other than Japan.

6. PLOS authors have the option to publish the peer review history of their article (what does this mean?). If published, this will include your full peer review and any attached files.

Reviewer #1: **Yes: **Akihiro Nomura, MD, PhD

Reviewer #2: No

---

## [Author Response · Author response to Decision Letter 0]

11 Nov 2020

Reviewer 1,

We wish to express our appreciation to the reviewers for their insightful comments on our paper. The comments have helped us significantly improve the paper. 

Major comments:

Comment 1:

Please clarify the benefit of using the TCI instead of the FTND-based general craving score in this study (e.g. easy to use, less time to fill them out…?).

Response 1:

We really thank the reviewer for pointing out this very important issue. FTND is a measure of the level of physical nicotine dependence and it associated with the degree of craving. In Japan, the Japanese-translated version of the 10-item Questionnaire on Smoking Urges-Brief (QSU-brief) is the only available measure of craving. However, the QSU-brief may be too long to use in clinical settings. Our new craving index, TCI, consists of only two axes; therefore, the TCI is a useful indicator to assess tobacco craving in patients and requires less time to fill out the medical questionnaire. 

 According to the comment, we added the following sentence in the Discussion P18, L1: “As the TCI questionnaire consists of only two questions, patients can easily fill it out. Therefore, we think that it is a useful questionnaire that can be administered in various clinical settings for the assessment of tobacco craving.”

Comment 2:

How many patients received smoking cessation medications as nicotine patch or varenicline? Since these medications can ameliorate the cravings, low prescription rate may affect the results.

Response 2:

We greatly appreciate the reviewer for this comment. As the reviewer said, prescription may affect the results. According to the comment, we added an explanation of prescription in Methods P7, L1: “patients were given a prescription for either varenicline or nicotine patches.” In addition, we added the numbers of patients who received a prescription of varenicline or nicotine patches in Table 1. Also, we added prescription (nicotine patches/ varenicline) as a confounding factor in logistic regression analysis in Table 2. According to this revision, we re-calculated the odds ratios and wrote the new odds ratios in the Abstract and Results as follows: “adjusted odds ratio: 2nd session: 0.30, 3rd session: 0.15, 4th session: 0.06, 5th session: 0.02”

Comment 3:

Table 1. What is “cohabiter”? Please clarify and explain it.

Response 3:

We changed the word “cohabiter” to “cohabitant” in Table 1. A cohabitant was defined as a person living together with the participant of the SCT. We added the definition of cohabitant in the Methods section and in a note under Table 1.

Comment 4:

Table 1, last 2 rows. It seems that 2 tables were mixed, “N (%)” and “mean, SD”, very confusing. Please separate them or change the style.

Response 4:

We sincerely appreciate the reviewer’s important comment. We changed Table 1 according to the reviewer’s comment. 

Comment 5:

How do you think we can apply these results into the smoking cessation program? Should we aggressively intervene cravings at 2nd to 4th sessions for smoking cessation success?

Response 5:

We really thank the reviewer for pointing out this very important issue. As the reviewer said, it is important to describe how to apply these results for Japanese SCT. According to the comment, we added a sentence in the Discussion section as follows: (P15, L17) “Therefore, we think that the TCI is a useful indicator for monitoring the efficacy of success of quitting smoking during smoking cessation treatment.”

Minor comments:

Comment 1:

Please remove abbreviation “(TCI)” from the title.

Response 1:

According to the comment, we removed abbreviation “TCI” from the title.

Comment 2:

In conclusion, you cannot say definitive things from the study. Change the last sentence as “These findings indicate that … smoking cessation interventions might be useful …”.

Response 2:

According to the comment, we made the following change in the sentence on P 18, L4: “These findings indicate that monitoring the TCI of participants during smoking cessation intervention might be useful when considering appropriate medication and counseling methods.”

Again, thank you for giving us the opportunity to strengthen our manuscript with your valuable comments and queries. We have worked hard to incorporate your feedback and hope that these revisions persuade you to accept our submission.

Reviewer 2,

We wish to express our appreciation to the reviewers for their insightful comments on our paper. The comments have helped us significantly improve the paper. 

Comment 1:

Methods Settings and participants: The authors mention “We selected 889 participants who answered the TCI questionnaire in first session of the SCT.” I wonder how to select this study population. Did the authors offer Tobacco Craving Index questionnaire to all participants in the smoking cessation therapy? Did these 889 participants who answered the TCI questionnaire in first session of the SCT complete all 5 sessions of Japanese smoking cessation therapy?

Response 1:

We sincerely appreciate the reviewer’s important comment. According to the comment, we added the following sentence in the Methods section P6, L15: “The TCI questionnaire was administered to each patient at every session of the SCT starting in October 2009. The subjects of the present study were 889 patients who gave written informed consent to receive Japanese SCT and who answered the TCI questionnaire in the first session of the SCT between October 2009 and June 2014.”.

This study included all participants who answered the TCI questionnaire at the first session of SCT. Participants who did not complete all five sessions of the SCT were considered to have failed smoking cessation in this study. According to the reviewer’s comment, we added the following sentence on P8, L3: “Those who dropped out of the SCT from the second to the fifth session were defined as having failed quitting smoking.”

Comment 2:

Please describe medical treatment in the smoking cessation therapy, that is, transdermal nicotine patches or varenicline. Prescribed pharmacotherapy might be associated with success of quitting smoking as a confounding factor. 

Response 2:

We greatly appreciate the reviewer for this comment. As the reviewer stated, prescription may affect the results. According to the comment, we added an explanation about the prescription in the Methods P7, L1: “patients were given a prescription for either varenicline or nicotine patches.” In addition, we added the numbers of patients who received varenicline or nicotine patches in Table 1. Also, we added prescription (nicotine patches/ varenicline) as a confounding factor in logistic regression analysis in Table 2. According to this revision, we re-calculated the odds ratios and wrote the new odds ratios in the Abstract and Results as follows: “adjusted odds ratio: 2nd session: 0.30, 3rd session: 0.15, 4th session: 0.06, 5th session: 0.02”

Comment 3:

The authors should mention that advantage of their Tobacco Craving Index questionnaire compared with other tools to evaluate nicotine craving. In addition, I think their Tobacco Craving Index questionnaire should be validated in other countries other than Japan.

Response 3:

We sincerely appreciate the reviewer’s important comment. We already mentioned the advantage of TCI in the Introduction and Discussion (P4, L11, P15, L5). To further explain the advantage of TCI, we added the following sentence in the Conclusion (P18, L1): “As the TCI questionnaire consists of only two questions, patients can easily fill it out. Therefore, we think that it is a useful questionnaire that can be administered in various clinical settings for the assessment of tobacco craving.”

In addition, according to the reviewer’s comment, we added a sentence on P18, L6, as follows: “Further studies are needed to validate the TCI in countries other than Japan.”

Again, thank you for giving us the opportunity to strengthen our manuscript with your valuable comments and queries. We have worked hard to incorporate your feedback and hope that these revisions persuade you to accept our submission.

---

## [Decision Letter · Decision Letter 1]

20 Nov 2020

Relationship between the strength of craving as assessed by the Tobacco Craving Index and success of quitting smoking in Japanese smoking cessation therapy

PONE-D-20-25851R1

Dear Dr. Taniguchi,

We’re pleased to inform you that your manuscript has been judged scientifically suitable for publication and will be formally accepted for publication once it meets all outstanding technical requirements.

Kind regards,

Neal Doran

Academic Editor

PLOS ONE

Additional Editor Comments (optional):

Reviewers' comments:

Reviewer's Responses to Questions

**Comments to the Author**

1. If the authors have adequately addressed your comments raised in a previous round of review and you feel that this manuscript is now acceptable for publication, you may indicate that here to bypass the “Comments to the Author” section, enter your conflict of interest statement in the “Confidential to Editor” section, and submit your "Accept" recommendation.

Reviewer #1: All comments have been addressed

Reviewer #2: All comments have been addressed

2. Is the manuscript technically sound, and do the data support the conclusions?

Reviewer #1: Yes

Reviewer #2: Yes

3. Has the statistical analysis been performed appropriately and rigorously? 

Reviewer #1: Yes

Reviewer #2: Yes

4. Have the authors made all data underlying the findings in their manuscript fully available?

Reviewer #1: Yes

Reviewer #2: Yes

5. Is the manuscript presented in an intelligible fashion and written in standard English?

Reviewer #1: Yes

Reviewer #2: Yes

6. Review Comments to the Author

Reviewer #1: The authors sufficiently replied to my comments and edited the manuscript.

Reviewer #2: According to my comments, the authors appropriately revised the manuscript. I do not have any more comments.

7. PLOS authors have the option to publish the peer review history of their article (what does this mean?). If published, this will include your full peer review and any attached files.

Reviewer #1: **Yes: **Akihiro Nomura

Reviewer #2: **Yes: **Hiromi Tomioka

---

## [Editor Report · Acceptance letter]

26 Nov 2020

PONE-D-20-25851R1 

Relationship between the strength of craving as assessed by the Tobacco Craving Index and success of quitting smoking in Japanese smoking cessation therapy 

Dear Dr. Taniguchi:

I'm pleased to inform you that your manuscript has been deemed suitable for publication in PLOS ONE. Congratulations! Your manuscript is now with our production department. 

Kind regards, 

on behalf of

Dr. Neal Doran 

Academic Editor

PLOS ONE